# Functional analysis of the nonstructural protein NSs of tomato zonate spot virus

Jing Li[1]☉, Si Chen[1,2]☉, Run-Shuang Qiu[2], Li-Zhen Zhang[2], Yue Chen[2], Xue Zheng[2], Ting-Ting Li[2], Li-Hua Zhao📧[2]*, Zhong-Kai Zhang[2]*

1 Life Science College, Southwest Forestry University, Kunming, China, 2 Yunnan Provincial Key Lab of Agricultural Biotechnology, Key Lab of Southwestern Crop Gene Resources and Germplasm Innovation, Ministry of Agriculture, Institute of Biotechnology and Germplasm Resources, Yunnan Academy of Agricultural Sciences, Kunming, China

☉ These authors contributed equally to this work.
* 356429686@qq.com (LHZ); zhongkai99@sina.com (ZKZ)

**Data Availability Statement:** All relevant data are within the paper and its Supporting Information files.

**Funding:** Zhong-kai Zhang National Natural Science Foundation of China (No. U1802235) Li-

## Abstract

Tomato zonate spot virus (TZSV), a member of the genus *orthotospovirus*, causes severe damage to vegetables and ornamental crops in southwest China. The NSs protein is an RNA silencing suppressor in various *orthotospovirus* like TZSV, but its mechanism and role in virus infection are poorly understood. Here, we observed that an NSs-GFP fusion protein was transiently expressed on the plasma membrane and Golgi bodies in *Nicotiana benthamiana* plants. The TZSV *NSs* gene was silenced and infiltrated into *N. benthamiana* and *N. tabacum* cv. K326. RT-qPCR and Indirect enzyme-linked immunosorbent assay (ID-ELISA) showed that the transcription and the protein expression of the *NSs* gene were inhibited by more than 90.00%, and the symptoms on silenced plants were alleviated. We also found that the expression of the *Zingipain-2-like* gene significantly decreased when the *NSs* gene was silenced, resulting in co-localization of the NSs-GFP and the Zingipain-2-like-mCherry fusion protein. The findings of this study provide new insights into the mechanism of silencing suppression by NSs, as well as its effect on systemic virus infection, and also support the theory of disease resistance breeding and control and prevention of TZSV in the field.

## Introduction

Tomato zonate spot virus (TZSV), the dominant species, which belongs to the *orthotospovirus* genus of the *Bunyaviridae* family in Yunnan Province, is transmitted by thrips [1]. In recent years, TZSV has been prevalent in Yunnan, Guizhou, Guangxi, and other regions of China and Southeast Asia. TZSV infection is often associated with severe disease symptoms, including concentric rings and spots on fruits and necrosis of leaves. TZSV has a very broad host range, infecting more than 20 types of economically important crops and weed species, which belong to 7 families [2, 3]. This prevalence has not only led to production losses and quality problems for important vegetable and ornamental plants in Southwest China but also seriously threatened economically important crops for local farmers [4].

hua Zhao National Natural Science Foundation of China(No.31960531), Yunnan Fundamental Research Projects (202101AT070269).

**Competing interests:** The authors have declared that no competing interests exist.

TZSV consists of spherical, enveloped particles that become distributed in the cytoplasm and the endoplasmic reticulum in mesophyll cells [1, 5]. Like other members of the genus *orthotospovirus*, TZSV contains a tripartite genome consisting of large (L), medium (M), and small (S) segments. The L RNA segment is negative and encodes the RNA-dependent RNA polymerase (RdRp). The M RNA segment encodes the nonstructural NSm protein and the viral glycoprotein precursor (Gn/Gc), while S RNA encodes the nonstructural NSs protein and the nucleocapsid protein N, and the N protein is responsible for the envelope formation of the viral genome RNA [6, 7].

The NSs protein is an RNA-silencing suppressor encoded by orthotospoviruses, such as TZSV, tomato spotted wilt orthotospovirus (TSWV), groundnut ringspot virus (GRSV), and tomato yellow ring virus (TYRV) [1, 8, 9]. The silencing suppressor of TSWV is the multifunctional NSs protein that is necessary for systemic movement in plants and can influence the emission of plant volatiles and suppress the JA-regulated plant defenses, resulting in the enhanced attractiveness of plants to flower thrips (*Frankliniella occidentalis*). NSs can also functionally replace potyviral HC-Pro and promote systemic infection and symptom development by suppressing antiviral RNA silencing [6, 10–12]. The NSs protein of watermelon silver mottle virus (WSMoV) has the function of transmission of the virus by *T. palmi* [13]. The NSs protein of TZSV activates a hypersensitive response in resistant plants and could interact with the VDAC protein in F. *occidentalis*, regulating the transmission of the persistent-propagative plant viruses [14]. The mechanisms of RNA-silencing suppression in TZSV by NSs and its role in virus infection need further exploration.

Virus-induced gene silencing (VIGS) is a method to study the functions of plant and pathogen genes by the agroinfiltration or biolistic inoculation of plants. VIGS has been successfully used to investigate gene function and disease resistance. Silencing of the *LeCTR1* gene in tomatoes led to an accumulation of ROS and increased the expression of *NPR1*, *PR1*, *PR5*, and *AOS2* genes to prevent the *tomato leaf curl virus* (ToLCV) infection [15]. When the *H2B* and *CoiI* genes in *N. benthamiana* were silenced, the contents of the phytohormone salicylic acid (SA) and jasmonic acid (JA) increased, and the infection with potato virus X (PVX) and (TSWV) was inhibited [16, 17]. Macharia et al. found that silencing of the *NbHYPK* and *ATG8* genes could enhance autophagocytosis and help combat the TMV infection [18]. The *N* gene of TSWV was inserted into the TRV vector to further study the gene function, but there have been no studies on inserting TZSV genes into VIGS vectors [19]. In this study, *NSs* gene was silenced by constructing the TRV-pTV00 vector and then infiltrated into *N. benthamiana* and *N. tabacum* cv. K326. RT-qPCR and ID-ELISA assays showed that *NSs* gene transcription and protein expression were inhibited more than 90.00%, and the symptoms in the silenced plants were alleviated. We also observed that the expression of the *Zingipain-2-like* gene significantly decreased when *NSs* gene was silenced. Furthermore, the results showed the co-localization expression of NSs-GFP and Zingipain-2-like-mCherry fusion protein on the plasma membrane in *Nicotiana benthamiana* plants. This is the first report using the TRV VIGS system to analyze the functions of the TZSV gene, which have important new implications for mechanistic studies of the suppression of gene silencing by NSs and their effects on systemic infection by the virus.

## Materials and methods

### Materials

The pTV00, pBINTRA, and pTV00-PDS, pCAMBIA-GFP, pBI121-mCherry vectors were provided by Professor Jianqiang Wu's laboratory at the Kunming Institute of Botany, Chinese Academy of Sciences (KIB CAS) [20]. The TZSV YN-Chili isolate was collected from the

infected tomato field in Yuanmou, Yunnan Province, China, and maintained on *N. benthamiana* [1]. *N. tabacum* cv. K326 and *N. benthamiana* were cultivated at the Yunnan Academy of Agricultural Science. Primers were designed using the Primer 5 Design Program based on the sequences of TZSV *NSs* gene and *Zingipain-2-like* gene published in the NCBI database (registration number: EF552433.1, LOC107763929) (S1 Table).

## TZSV inoculation

The frictional inoculation method was used to artificially infect *N. tabacum* cv. K326 and *N. benthamiana* plants at the six-leaf stage. TZSV-infected *N. benthamiana* leaves were homogenized in the PBS buffer (100 mg/mL) containing 137 mM NaCl, 1 mM $KH_2PO_4$, 8 mMNa$_2$H-PO$_4$·12H$_2$O, and 3 mM KCl, and thereafter the homogenate was applied uniformly to 3 leaves per plant (1 mL). Ten minutes after the TZSV inoculation, the inoculated plant leaves were rinsed with ddH$_2$O. Plant leaves inoculated with PBS buffer were used as controls, and five replicates were used for each sample. Five days after inoculation, symptoms appeared.

## RNA extraction and RT-PCR

According to the instructions provided in the RNA extraction kit (Roche, America), the total RNA of plant leaves was extracted. The first-strand cDNA synthesis kit was used for reverse transcription (TransGen, China). Q5 High-Fidelity DNA Polymerase (NEB, England) was used to amplify the target fragments. The PCR solution consisted of 1 μL of cDNA, 10 μL of 10× EasyTaq Buffer, 1 μL of 2.5 mM dNTPs, 2.5 μL of forward and reverse primers, and 0.5 μL of Q5 DNA polymerase; ddH$_2$O was added to the solution to obtain the final volume of 50 μL. The PCR conditions were as follows: 35 cycles of denaturation at 98˚C for 40 s, 98˚C for 10 s, 60˚C for 20 s, and 72˚C for 30 s, followed by 72˚C for 2 min. The amplified products were subsequently analyzed using a UVP gel-imaging system.

## Plasmid constructs

(i) PCR was used to amplify the desired fragments with specific primers (S1 Table) using cDNA prepared from the plant tissues inoculated with TZSV. The target PCR fragments were excised, and DNA was extracted using an appropriate kit (AxyGen, America). The amplified fragments of the *NSs* gene and pTV00 vectors were digested by the restriction enzymes *BamHI* and *HindIII* according to the manufacturers' instructions, and the purified products of the *NSs* sequence were inserted into the pTRV-pTV00 vectors using T4 DNA ligase (NEB, England). The fragments of the to-be-silenced *NSs* gene were cloned in the pTV00 vectors. The vectors were then transformed into competent cells of *E. coli* strain DH5α. The selected positive clones were transferred to Guangzhou Huada Gene Technology Co., Ltd., for sequencing. Plasmids with the correct sequence were used to transform *A. tumefaciens* electrocompetent cells.

(ii) The sequences of *NSs* and *Zingipain-2-like* genes were amplified from the total RNA isolated from tobacco plants infected with TZSV using reverse transcription-PCR (RT-PCR) and the special primers (S1 Table). The PCR fragments of *NSs* and *Zingipain-2-like* genes were digested with endonuclease and inserted into vector of pCAMBIA-GFP and pBI121-mCherry to obtain pCAMBIA-NSs -GFP and pBI121-Zingipain-2-like-mCherry, respectively. The Golgi marker ST52-mCherry was amplified from the total RNA isolated from both tobacco and Arabidopsis [21].

## Infiltration of the VIGs vector in tobacco leaves

The pTRV-pTV00-NSs constrcture was transformed into *Agrobacterium tumefaciens* strain GV3101 by electroporation. For every 1 mL of the inoculation solution, 2.5 mL of the liquid YEP culture plus 2.5 mL of pBINTRA liquid YEP culture were used. For each construct, the final inoculation solution was made by mixing equal volumes of the resuspended *Agrobacterium* carrying the pTRV-pTV00-NSs construct and pBINTRA.

Inoculation was performed using a 1-mL syringe, and the inoculation solution of 1 mL was pressure injected into individual leave and per plant was injected 3 leaves. Inoculation of pTRV-pTV00-NSs construct was performed using a 1-mL syringe, and the inoculation solution of 1 mL was pressure injected into individual leave and per-plant was injected 3 leaves. Phytoene dehydrogenase gene (PDS) as an indicator gene, the plant leaves will turn white when it was silenced, so the leaves injected with pTRV-pTV00-PDS construct becoming bleached at approximately 10–14 days, the leaves performed with pTRV-pTV00-NSs construct inoculated with TZSV. Plants injected with pTRV-pTV00 and inoculated with TZSV alone served as positive controls. At approximately 5–10 days post-inoculation, samples were collected and healthy plants were used as blank controls.

Plants injected with pTRV-pTV00 and inoculated with TZSV alone served as positive controls. At approximately 5–10 days post-inoculation, samples were collected and healthy plants were used as blank controls.

## Confocal laser scanning microscopy and co-localization

The leaf epidermis was dissected from the areas of the agroinfiltrated *N. benthamiana* leaves and placed in water between two coverslips. The confocal images were captured with the inverted TCS SP8 and 10× water immersion objective lenses. GFP was excited at a wavelength of 488 nm, and emissions were captured at 497–520 nm. Moreover, mCherry was excited at a wavelength of 561 nm and emissions were captured at 585–615 nm. Images were processed using the TCS SP8 and Adobe (San Jose, CA, USA) Photoshop.

## RT-qPCR

The sequences of the *NSs* gene were inserted into the pMD18-T vector, and the plasmids were diluted 10-fold with a gradient to obtain plasmids with 10-fold dilution series ($10^{-1}$,$10^{-2}$,$10^{-3}$,$10^{-4}$,$10^{-5}$,$10^{-6}$), and then used as calibrators to construct a standard curve. The reaction system consisted of the components including 1 μL of cDNA, 2.5 μL of forward and reverse primers,1 0 μL of FastStart Universal SYBR Green Master (Rox), and 4 μL of ddH$_2$O. The reaction conditions were as follows: initial denaturation step at 95˚C for 10 min, followed by 40 cycles of 95˚C for 10 s, 57˚C for 30 s, and 72˚C for 30 s.

## ID-ELISA

The contents of the proteins were tested by ID-ELISA according to the instructions for antibodies to determine the antiviral activity of VIGS. Leaves (0.2 g) were homogenized using a mortar and pestle and diluted 1: 3 in a PBS buffer. Crude extracts (100 μL) were added into ELISA plate wells and incubated at 37˚C for 2 h. The plate was then washed with PBST buffer. TZSV NSs rabbit antibodies were diluted in a conjugation buffer, and afterward, 100 μL of goat anti-rabbit IgG-AP conjugate (Sigma, USA) was added to each well. The color-developing solution was dissolved in p-nitrophenyl phosphate disodium hexahydrate (Sigma-Aldrich) in substrate buffer to obtain a final concentration of 1 mg/mL. The absorbance was determined at 405 nm using the ELx 808 microplate ELISA reader (Bio-Tek, USA). Healthy leaves were

used as negative controls, and TZSV-infected leaves were used as positive controls. The PBS buffer was used as a blank control.

## Results

### NSs was localized to the PM and Golgi bodies

Previous studies have shown that confocal laser scanning microscopy (CLSM) was used to analyze the protein localized in living cells [22]. *N. benthamiana* is also a model plant species to assess the subcellular localization of viral proteins. To characterize the subcellular targeting of the NSs in plant cells, the recombinant NSs-GFP (green fluorescent protein) was first transiently expressed in leaf epidermal cells of *N. benthamiana* by agroinfiltration. NSs-GFP was then detected to be associated with the plasma membrane structures by CLSM (Fig 1A–1C). To determine whether the NSs-GFP bodies are colocalized with the Golgi stacks, we also checked the localization of NSs-GFP for Golgi bodies using the marker ST52-mCherry [23]. At 48 h after agroinfiltration on *N. benthamiana*, we found that the NSs-GFP bodies are colocalized with the Golgi stacks (Fig 1D–1F), suggesting that the NSs-GFP protein was targeted to the Golgi apparatus.

### Silencing of the *NSs* gene

TRV vector was widely used to study the interactions between viruses and hosts and the functions of plant genes [24]. In the present study, specific primers (TZVNSsF1/TZVNSsR1)

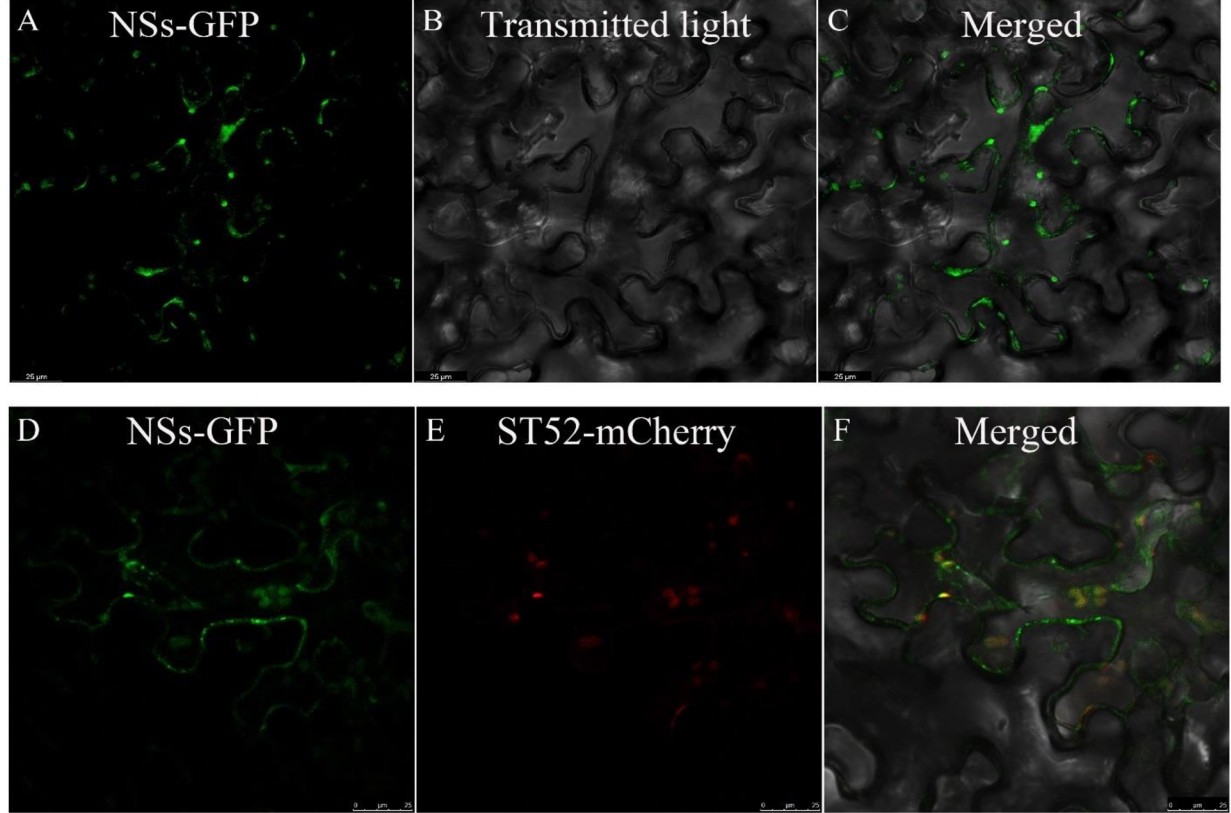

**Fig 1. The NSs protein in TZSV is localized at the PM and Golgi bodies.** A-C: localization of NSs-GFP at 48 h post infiltration (hpi). Bar, 25 μm. D-F: co-localization of NSs-GFP with Golgi bodies at 48 hpi. Bar, 25 μm. The fluorescence derived from N. *benthamiana* leaves was monitored using a confocal Leica TCS SP8.

containing *BamHI* and *HindIII* restriction enzyme recognition sites were used to amplify the *NSs* gene fragments (S1 Fig), and the DNA was inserted into the pEASY-Blunt-Zero vector (TransGen, Beijing) for sequencing to ensure that the base sequences were not mutated.

The fragments of the *NSs* gene were amplified and inserted into the pEASY-T1 Simple vector (Trans, Beijing). The concentration and the $OD_{260/280}$ value for the recombinant plasmid containing the *NSs* gene were 225.82 ng/μL and 1.83, respectively. The plasmid DNA with gradient dilutions of $10^{-1}$ to $10^{-6}$ was used as a template. Standard curves and amplification curves of RT-qPCR data for the *NSs* gene were obtained by automatic analysis performed by the software system. The standard curve equation of the *NSs* gene was Y = -3.444X+34.42, and the amplification efficiency and the correlation coefficient were 90.90% and 0.998, respectively. The results showed that the plasmid DNA could be used as a calibration product to determine the copy numbers of the gene.

The *PDS* gene was used as a positive control to ensure the success of silencing. The leaves of *N. benthamiana* without this gene were bleached. However, there were no phenotypic changes in leaves of the *N. tabacum* cv. K326, despite the fact that these species belong to the same genus. TZSV was inoculated on the two species tobaccos that the plants were injected with pTRV-pTV00-NSs construct and positive control plants (inoculated only with TZSV), respectively, when the leaves in the veins of *N. benthamiana* turned white injecting with pTRV-pTV00-PDS construct. After 5 days of inoculation with TZSV, the shrinkage also appeared on the leaves of *N. benthamiana* plants injected with the pTRV-pTV00-NSs vector before inoculation with TZSV (Fig 2A). Severe leaf shrinkage also occurred in positive control plants (inoculated only with TZSV) (Fig 2B); however, for *N. tabacum* cv. K326, the symptoms in the VIGS plants and positive control plants were not different. To further determine the effects of the *NSs* gene, its transcription level was measured by RT-qPCR and found to be significantly decreased compared to that of the positive control (inoculated only with TZSV) in both hosts. The silencing efficiencies of the TZSV *NSs* gene were 99.16% in *N. benthamiana* and 92.24% in *N. tabacum* cv. K326 (Fig 3A and 3B). The results indicate that the pTRV-pTV00-NSs VIGS vector was successfully constructed, and the *NSs* gene might be associated with TZSV infection.

## Inhibition of the NSs protein expression

The NSs protein levels were measured by ID-ELISA at 3, 5, 7, and 9 days post-inoculation with TZSV, and found to be significantly decreased in leaves of *N. benthamiana* and *N. tabacum* cv.

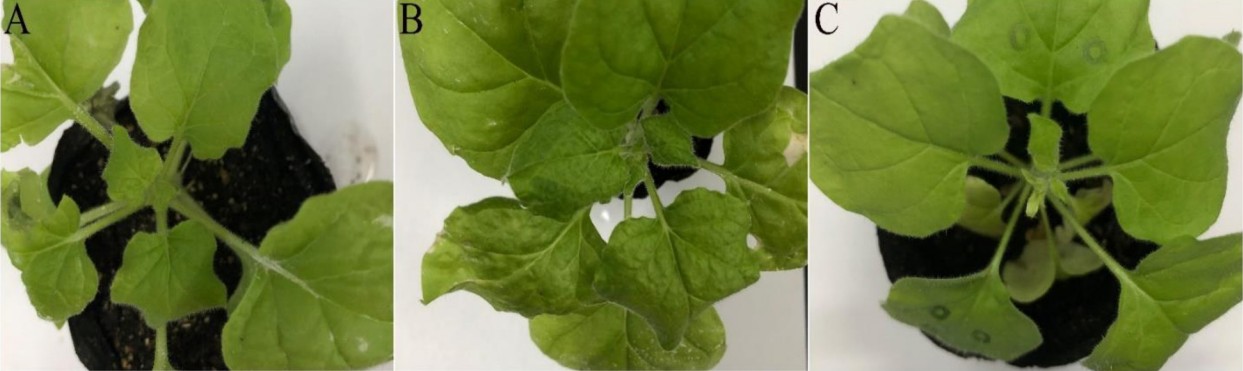

**Fig 2. The symptom of TZSV infection on *N. benthamiana*.** A: infiltrating pTRV-PTV00-NSs construct in the leaves and inoculated with TZSV; B: positive control (inoculated with TZSV only); C: negative control (the healthy plant).

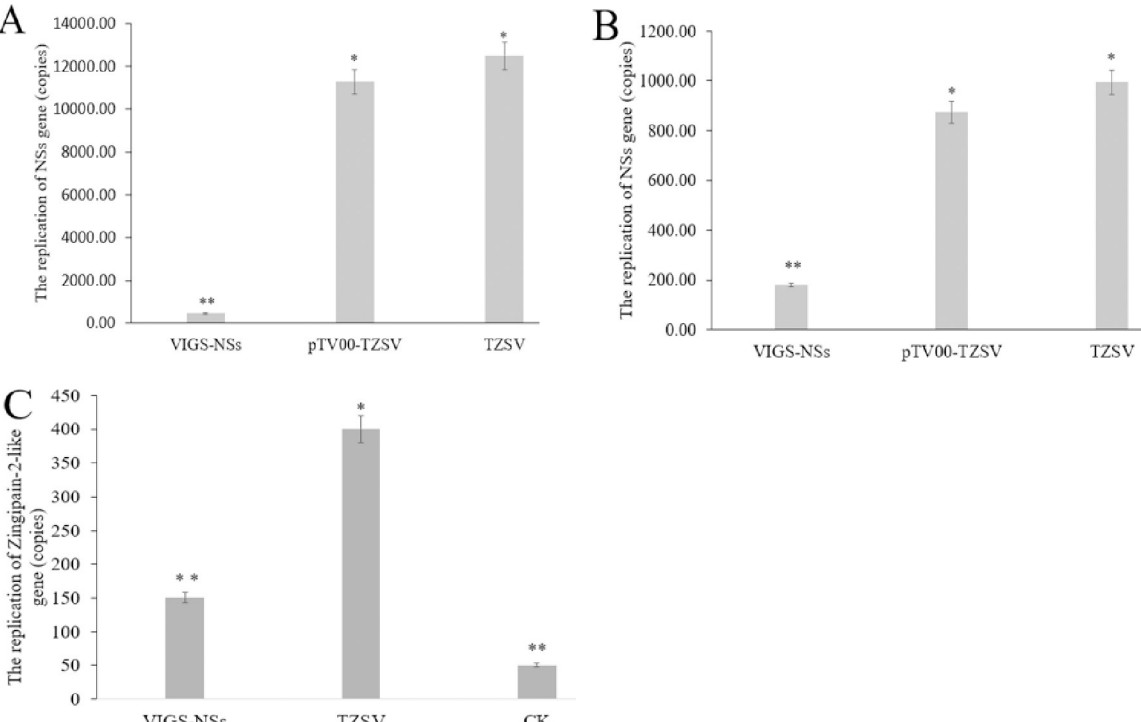

**Fig 3. RT-qPCR detected the transcription of *NSs* and *Zingipain-2-like* genes after *NSs* gene silenced.** A: the expression of *NSs* gene is detected with different treatment in *Nicotiana Benthamiana* leaves; B: the expression of *NSs* gene is detected with different treatment in *N. tabacum* cv. K326 leaves; C: the expression of *Zingipain-2-like* gene is detected in *Nicotiana Benthamiana* leaves; VIGS-NSs: infiltrated pTRV-pTV00-NSs construct and inoculated with TZSV; pTV00-TZSV: infiltrated pTRV-pTV00 vector and inoculated with TZSV; TZSV: positive control and only inoculated with TZSV; CK: negative control (the healthy plant). All values are means ± SE. *means differences are significantly different at P ≤ 0.05.

K326 plants treated with the pTRV-pTV00-NSs construct and inoculated with TZSV compared to those in positive plants at 7 and 9 days (Table 1). The results showed that the expression of the *NSs* protein in both *N. benthamiana* and *N. tabacum* cv. K326 hosts was inhibited.

## The dependence of *NSs* gene silencing on the *zingipain-2-like* gene

The *Zingipain-2-like* gene was a homocysteine protease, which possesses cysteine-type endo-peptidase activity and participates in the regulation of plant-type hypersensitive response [25, 26]. To confirm whether the host factor for the *Zingipain-2-like* gene made closely relationship with RNA silencing suppressing by the NSs gene of TZSV, the expression of the *Zingipain-2-like* gene performed by *NSs*, the silencing suppressor in plants, was detected by RT-qPCR assay, and found to be significantly decreased in the *NSs* gene-silenced plants compared to positive plants, compared with the CK, expression of the Zingipain-2-like gene was up-regulated in both positive and NSs-silenced plants infected by TZSV plants, but higher in the positive plant. The rationale behind these data could be that *Zingipain-2-like* gene expression was induced by NSs (Fig 3C).

To further investigate whether NSs and the Zingipain-2-like protein were colocalized and to identify the specific structures in living cells, we transiently expressed the recombinant NSs-GFP (green fluorescent protein) and Zingipain-2-like-RFP (red fluorescent protein) in leaf epidermal cells of *N. benthamiana* by agroinfiltration. The co-expression of NSs-GFP with the Zingipain-2-like-RFP confirmed the co-localization of NSs and Zingipain-2-like at the plasma

**Table 1. Detection of NSs protein content with different treatment in both host using ID-ELISA.**

| Host species | pTRV-pTV00-NSs+TZSV | | | | TZSV | | | | pTV00+TZSV | | | | CK | | | |
|---|---|---|---|---|---|---|---|---|---|---|---|---|---|---|---|---|
| | 3 d | 5 d | 7 d | 9 d | 3 d | 5 d | 7 d | 9 d | 3 d | 5 d | 7 d | 9 d | 3 d | 5 d | 7 d | 9 d |
| *N. benthamiana* | 0.207 ± 0.010a | 0.214 ± 0.055a | 0.244 ± 0.019a | 0.201 ± 0.011a | 0.219 ± 0.001ac | 0.289 ± 0.002ac | 0.446 ± 0.008bc | 0.369 ± 0.014bc | 0.208 ± 0.010ac | 0.287 ± 0.005ac | 0.396 ± 0.002bc | 0.379 ± 0.017bc | 0.169 ± 0.009d | 0.163 ± 0.013d | 0.168 ± 0.014d | 0.162 ± 0.013d |
| *N. tabacum* cv. K326 | 0.181 ± 0.009a | 0.218 ± 0.007a | 0.257 ± 0.006a | 0.225 ± 0.062a | 0.218 ± 0.006ac | 0.282 ± 0.008ac | 0.434 ± 0.043bc | 0.391 ± 0.011bc | 0.217 ± 0.007ac | 0.245 ± 0.002ac | 0.374 ± 0.010bc | 0.368 ± 0.001bc | 0.163 ± 0.006d | 0.147 ± 0.007d | 0.163 ± 0.004d | 0.133 ± 0.008d |

pTRV-pTV00-NSs +TZSV: infiltrated pTRV-pTV00-NSs construct and inoculated with TZSV; TZSV: positive control and means only inoculated with TZSV; pTV00-TZSV: means infiltrated pTRV-pTV00 vector and inoculated TZSV; CK: negative control and means the leaves with no treatment. All values are means ± SE. Means in a column followed by different letters are significantly different at P ≤ 0.05.

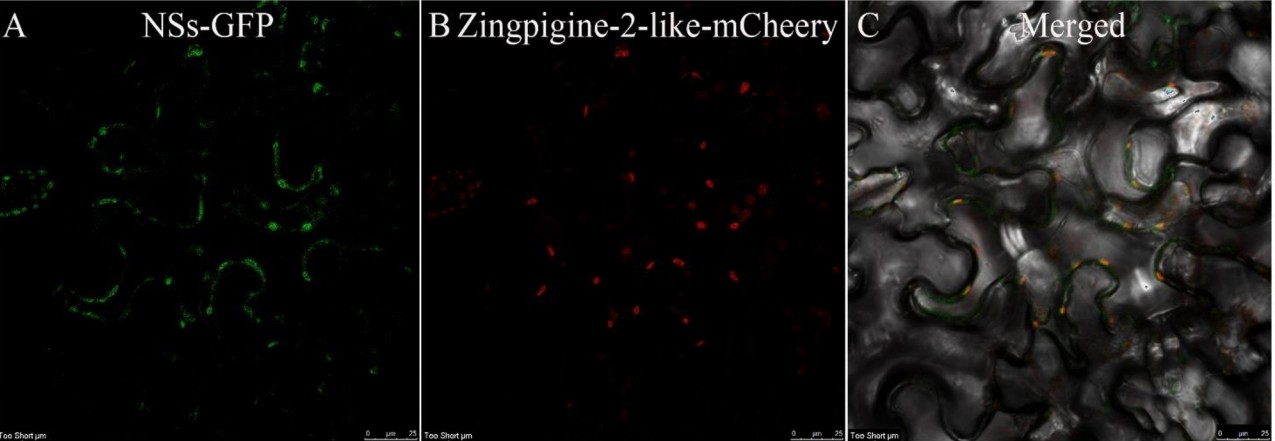

**Fig 4. The NSs protein in TZSV is co-localized with the Zingipain-2-like protein.** A-C: Colocalization of NSs-GFP with Zingipain-2-like at 48 hpi. Bar, 25 μm. The fluorescence derived from N. *benthamiana* leaves was monitored using a confocal Leica TCS SP8.

membrane by the confocal laser scanning microscope (CLSM) (Fig 4A–4C). Our results clarified that expression of *NSs* is closely correlated to that of Zingipain-2-like when TZSV infected the plants.

## Discussion

The importance of NSs for tospoviral infection in plants was first discovered in the early 1990s. A higher virulence of TSWV isolates and more severe symptoms were observed with the elevated levels of NSs expression [27]. The TSWV NSs protein acts as a suppressor of RNA silencing through binding small and long dsRNA and suppresses short and long-distance viral accumulation and movement [28]. It also represents the Avr factor of the *Tsw* resistance gene in pepper [29, 30]. NSs of TSWV directly interact with *MYC2*, a regulator of the JA signaling, to disable JA-mediated activation of terpene synthase genes and attenuate host defenses, increasing the attractiveness of the plants to thrips, and thus transmitting the disease [31]. So far, the ability of NSs of TSWV to counteract defense mechanisms mediated by RNA silencing in plants has been demonstrated; however, the mechanism of RNA silencing suppression by NSs in TZSV and its role in virus infection are not yet clear. In the present study, the accumulation of the virus decreased and symptoms were alleviated when the *NSs* gene was silenced. The expression of the *Zingipain-2-like* gene was found to be significantly decreased in the *NSs* gene-silenced plants compared to positive plants, compared with the CK, expression of the Zingipain-2-like gene was up-regulated in both positive and NSs-silenced plants infected by TZSV plants, but higher in the positive plant. We first revealed that the *Zingipain-2-like* gene might be associated with this function.

The innate immune system of plants has two different layers, including microbe-associated (MAMP) or pathogen-associated molecular pattern (PAMP)-triggered immunity (PTI) and effector-triggered immunity (ETI). PTI is mediated by the corresponding membrane-anchored pattern recognition receptors (PRRs) in plants, which serve as the first line of defense against the pathogen. Many plant viruses and their encoded proteins that could inhibit PTI, like the NSs protein of TSWV that suppressed the production of reactive oxygen species (ROS), have been reported [30, 32, 33]. In this study, we found that NSs and Zingipain-2-like were colocalized at the plasma membrane; the expression of the *Zingipain-2-like* gene significantly decreased in *NSs* gene-silenced plants compared to positive plants, and Zingipain-2-like

took part in the regulation of hypersensitive response, suggesting that NSs and Zingipain-2-like might be associated with the activation of PTI-like responses.

In this study, *N. benthamiana* was used as a model plant, and the ability of the NSs protein of TZSV to target Golgi bodies in plant cells was observed for the first time. The targeting of the virus glycoproteins to the Golgi apparatus plays a pivotal role in the formation of enveloped spherical particles of the viruses belonging to the Bunyaviridae family [34–36]. However, the reason for the facilitation of the formation of enveloped spherical particles by the NSs protein remains to be extensively investigated.

In this study, TRV vectors were used to construct VIGS vectors of the *NSs* gene for the analysis of their functions. The results of RT-qPCR, as well as the plant disease symptoms, showed that the gene replication was inhibited up to 90%. ID-ELISA showed that the protein contents also significantly decreased. The high efficiency of gene silencing can be verified by sampling and testing immediately at the onset of the disease. At the same time, temperature also had effects on the silencing phenotypes in plants [37]. In this experiment, the temperature was strictly controlled, and thus, the gene was silenced at relatively high levels, and the duration was relatively long. In this study, the *N. benthamiana* leaves were bleached, but there were no phenotypic changes in leaves of *N. tabacum* cv. K326, despite the fact that both species belong to the same genus, indicating that the TRV-VIGS vector exhibited differing sensitivities to different host species. TRV has a wide range of hosts, with a significant difference in sensitivity to TRV between species and cultivars [38, 39]. For instance, TRV sensitivity testing was carried out on 21 gerbera cultivars, and the results revealed that only 5 cultivars showed photobleached PDS-silencing symptoms on newly developed leaves [40]. The VIGS method can be used for reverse genetics studies and the analysis of the functions of unknown genes.

In summary, our results revealed that NSs, a suppressor of RNA silencing in TZSV, was localized to the PM and Golgi bodies and might also be associated with Zingipain-2-like to activate PTI-like responses using VIGS and subcellular localization prediction.

## Supporting information

**S1 Fig. Amplication of the *NSs* gene sequence for VIGs.**
(TIF)

**S1 Table. Special primers.** The primers were used to construct the VIGS, fluorescence labeling, and RT-qPCR vectors.
(DOCX)

## Acknowledgments

The authors would like to thank Professor Jianqiang Wu's laboratory KIB. CAS. for providing the VIGS vectors. We also thank the Yunnan Academy of Agricultural Sciences Yunnan Provincial Key Lab of Agricultural Biotechnology for equipment support of the confocal microscope. Jing-Li and Si-Chen contributed equally to this work.

## Author Contributions

**Conceptualization:** Li-Hua Zhao, Zhong-Kai Zhang.

**Data curation:** Li-Hua Zhao.

**Formal analysis:** Li-Hua Zhao, Zhong-Kai Zhang.

**Funding acquisition:** Zhong-Kai Zhang.

**Methodology:** Si Chen, Run-Shuang Qiu, Yue Chen, Ting-Ting Li.

**Resources:** Zhong-Kai Zhang.

**Supervision:** Li-Hua Zhao, Zhong-Kai Zhang.

**Validation:** Li-Hua Zhao, Zhong-Kai Zhang.

**Writing – original draft:** Jing Li, Si Chen, Li-Hua Zhao.

**Writing – review & editing:** Jing Li, Li-Zhen Zhang, Xue Zheng, Li-Hua Zhao, Zhong-Kai Zhang.

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
