## [Decision Letter · Decision Letter 0]

10 Sep 2021

PONE-D-21-26999Analysis function for a nonstructural NSs protein of tomato zonate spot orthotospovirusPLOS ONE

Dear Dr. Zhao,

Thank you for submitting your manuscript to PLOS ONE. After careful consideration, we feel that it has merit but does not fully meet PLOS ONE’s publication criteria as it currently stands. Therefore, we invite you to submit a revised version of the manuscript that addresses the points raised during the review process.

We look forward to receiving your revised manuscript.

Kind regards,

Sek-Man Wong

Academic Editor

PLOS ONE

Journal Requirements:

3. Please include a copy of Table 2 which you refer to in your text on page 9.

Reviewers' comments:

Reviewer's Responses to Questions

**Comments to the Author**

1. Is the manuscript technically sound, and do the data support the conclusions?

Reviewer #1: Partly

Reviewer #2: Partly

2. Has the statistical analysis been performed appropriately and rigorously? 

Reviewer #1: I Don't Know

Reviewer #2: Yes

3. Have the authors made all data underlying the findings in their manuscript fully available?

Reviewer #1: No

Reviewer #2: Yes

4. Is the manuscript presented in an intelligible fashion and written in standard English?

Reviewer #1: No

Reviewer #2: No

5. Review Comments to the Author

Reviewer #1: The manuscript, numbered PONE-D-21-26999, entitled ‘Analysis function for a nonstructural NSs protein of tomato zonate spot orthotospovirus’, indicates that the transiently expressed TZSV NSs-GFP fusion protein was co-localized with the Golgi apparatus and the endogenous Zingipain-2-like protein of Nicotiana benthamiana. Silencing of TZSV NSs gene significantly decreased virus replication and attenuated symptom development, and also downregulated the expression of Zingipain-2-like gene in the tested plants. Authors provide new insights into the role of the TZSV NSs protein; however, many problems can be found in the manuscript, including writing, conclusive evidence and significance. I encourage the manuscript to be resubmitted after comprehensive improvement. Here are my comments:

1. Virus taxonomy and writing of scientific name and virus name. TZSV should belong to Tomato zonate spot orthotospovirus species, Orthotospovirus genus, Tospoviridae family, Bunyavirales order. Authors can visit the ICTV website for information. For abbreviation, a virus name is required. Therefore, the use of ‘tomato zonate spot virus (TZSV)’ is recommended.

2. The text needs to be completely corrected. Too many spelling and grammatical errors can be found in the text. In addition, the citation of some references seems inappropriate.

3. The legends of tables and figures have to be clearly described for readers to understand. Additionally, miscitation of tables and figures can be found. For example, the primers are listed in Table S1, but the citation is in Table 1 (page 4, line 93, and page 5, line 129); Table 2 is quoted at page 9 (line 229), there is actually no Table 2!

4. What is ‘PDS’, ‘TCS SP8’, ’PM’, etc.? When describing an acronym for the first time, the full name must be provided.

5. The results, such as the quantification of NSs gene (standard curve and copy number, page 8, lines197-206) and the phenomenon and molecular evidence of PDS gene silencing in the tested plants (lines 207-223), should be illustrated in figures. Was PDS silencing suppressed by the TZSV NSs protein? It is not clear about the response to TZSV infection (or NSs protein) in the PDS-silenced tobacco plants! What is the point of this? In fact, I am confused about Fig. 2. I think the PDS silencing here is meaningless!

6. Authors must explain why the Zingipain-2-like gene is investigated? It is not even described in the M & M section! Logically, preliminary research should be performed to reveal the possible role of Zingipain-2-like gene in orthotospoviral (or TZSV) infection. Relevant information should be added.

7. The description of ‘replication’ of NSs gene or Zingipain-2-like gene is incorrect. It is should be ‘transcription’!

8. Table 1 can be illustrated as a figure, and Fig. S1 can be removed.

Reviewer #2: The manuscript (PONE-D-21-26999) describes the functional investigation of NSs gene of tomato zonate spot orthotospovirus. The authors showed that NSs protein (fused with GFP) localized in plasma membrane and Golgi bodies. They then constructed NSs-silenced tobacco plants by VIGS and then inoculated with tomato zonate spot orthotospovirus (TZSV). By monitoring the symptom expression and detection the NSs expression level, mild symptom was observed in N. benthamiana, but not in N. tabacum cv. K326. However, NSs was decreased by more than 90% in both plants. They also found that the expression of a host gene, Zingipain-2-like gene, seemed to be induced by the expression of NSs, and these two proteins co-localized in the cell. This was the first demonstration of the involvement of NSs in infection of TZSV. While the story is of interest, flaws need to be fixed before acceptance for publication.

1. Title: changed to “Functional analysis of the nonstructural protein NSs of tomato zonate spot orthotospovirus”

2. Abstract: needs to be rewritten after modification of the text.

3. Materials and Methods: materials and methods should be introduced in a logical manner, e.g., NSs gene amplification should go ahead of the construction of NSs gene silencing construct.

1) pCAMBIA-GFP, where was it obtained?

2) Vector and construct are different. pTRV-pTV00-NSs is a construct, not a vector.

3) Amount of inoculum used in leaf infiltration?

4) Lines 141-143: Bleaching of the leaves of the PDS-positive control occurred at approximately 10-14 days post-inoculation, and TZSV was inoculated. — what did the authors mean?

5) Line1 164-165: The content of the genes was tested by ID-ELISA according to the antibody instructions to determine the antiviral activity of the VIGS. — not genes but proteins?

4. Results

1) Line 197: The copies of NSs gene was determined by RT-qPCR. — for what purpose?

2) Line 212-217: the sentence needs to be reorganized, Fig.2A goes before Fig. 2B

3) Line 212: TZSV was inoculated after 5 days,…--- compared with line 142 “10-14 days”?

4) Line 235: “the replication of Zingipain-2-like gene…”, changed to expression of Zingipain-2-like gene. Also the title of Fig. 3 should be changed accordingly.

5) Line 234-238: Fig. 3C does not tell whether or not NSs interacts with Zingipain-2-like gene. The fact was that Zingipain-2-like gene expression was induced by NSs.

6) Data in Fig. 3C was inaccurately explained in the text, compared with the CK, expression of the Zingipain-2-like gene was up-regulated in both normal and NSs-silenced plants infected by TZSV, but higher in the normal plant (positive plant). The rationale behind these data could be that the NSs in the positive plant was higher than in the NSs-silenced plant.

7) Line 245: “RNA silencing suppressor by NSs”, change to RNA silencing suppressor NSs?

8) Line 245: NSs made closely relationship with Zingipain-2-like…?

9) Legend to Fig. 3C: CK was not mentioned. Was CK a healthy plant?

10) Fig. 4: what were the organelles where the florescent signals were present?

5. Discussion

1) Silencing of NSs seemed to have different impact on symptom expression on N. benthamiana and N. tabacum cv. K326. Why? – need a discussion.

2) Line 259-262: “In the present study, the accumulation of the virus decreased and the symptom alleviated when the NSs gene was silenced, and we first revealed Zingipain-2-like gene might be associate with this function”. – the data did not support this claim (see questions in Result, 5 and 6). The involvement of Zingipain-2-like gene in TZSV infection has to be verified in a well-defined study.

6. PLOS authors have the option to publish the peer review history of their article (what does this mean?). If published, this will include your full peer review and any attached files.

Reviewer #1: No

Reviewer #2: No

---

## [Author Response · Author response to Decision Letter 0]

23 Nov 2021

Thank you for considering our work and give us an opportunity. We modified the content following the editor and the reviewers' comments, and outline every change point by point.

---

## [Editor Report · Decision Letter 1]

26 Nov 2021

PONE-D-21-26999R1Functional analysis of the nonstructural protein NSs  of tomato zonate spot virusPLOS ONE

Dear Dr. Zhao,

Thank you for submitting your manuscript to PLOS ONE. After careful consideration, we feel that it has merit but does not fully meet PLOS ONE’s publication criteria as it currently stands. Therefore, we invite you to submit a revised version of the manuscript that addresses the points raised by the reviewers and incorporate them into the revised manuscript to improve its clarity.

We look forward to receiving your revised manuscript.

Kind regards,

Sek-Man Wong

Academic Editor

PLOS ONE

Journal Requirements:

Additional Editor Comments:

The authors have not adequately addressed all questions raised by the reviewers. Please look at the yellow highlights in the file PONE-D-21-26999_R1 WSM - 26 Nov 2021.pdf and respond accordingly. Thanks.
---

## [Author Response · Author response to Decision Letter 1]

30 Nov 2021

We have modified the content following the reviewers' comments, and outline every change point by point.

---

## [Editor Report · Decision Letter 2]

3 Dec 2021

PONE-D-21-26999R2Functional analysis of the nonstructural protein NSs  of tomato zonate spot virusPLOS ONE

Dear Dr. Zhao,

Thank you for submitting your manuscript to PLOS ONE. After careful consideration, we feel that it has merit but does not fully meet PLOS ONE’s publication criteria as it currently stands. Therefore, we invite you to submit a revised version of the manuscript that addresses the points raised during the review process.

We look forward to receiving your revised manuscript.

Kind regards,

Sek-Man Wong

Academic Editor

PLOS ONE

Journal Requirements:

Additional Editor Comments:

The authors still did not answer reviewers' questions I have highlighted to them. For example, "Authors must explain why the Zingipain-2-like gene is investigated?". Please look at my yellow highlights in my previous document and answer them point-by-point.
---

## [Author Response · Author response to Decision Letter 2]

15 Dec 2021

We have done a thorough revision and believe that the revised version has been substantially improved accordingly. Enclosed please find our point-by-point responses to all the comments and suggestions from the reviewers and editor. Line numbers correspond to the orgianl manuscript.

---

## [Editor Report · Decision Letter 3]

20 Dec 2021

Functional analysis of the nonstructural protein NSs  of tomato zonate spot virus

PONE-D-21-26999R3

Dear Dr. Zhao,

We’re pleased to inform you that your manuscript has been judged scientifically suitable for publication and will be formally accepted for publication once it meets all outstanding technical requirements.

Kind regards,

Sek-Man Wong

Academic Editor

PLOS ONE
---

## [Editor Report · Acceptance letter]

12 Jan 2022

PONE-D-21-26999R3 

Functional analysis of the nonstructural protein NSs of tomato zonate spot virus 

Dear Dr. Zhao:

I'm pleased to inform you that your manuscript has been deemed suitable for publication in PLOS ONE. Congratulations! Your manuscript is now with our production department. 

Kind regards, 

on behalf of

Dr Sek-Man Wong 

Academic Editor

PLOS ONE